# Reviewing progress in public involvement in NIHR research: developing and implementing a new vision for the future

Sophie Staniszewska,[1] Simon Denegri,[2] Rachel Matthews,[3] Virginia Minogue[4]

[1]Warwick Research in Nursing, Division of Health Sciences, Warwick Medical School, Division of Health Sciences, University of Warwick, Coventry, UK
[2]NIHR National Director for Patients, Carers and the Public UCL School of Life and Medical Sciences, London, UK
[3]National Institute for Health Research (NIHR) Collaboration for Leadership and Applied Health Research and Care (CLAHRC), Northwest London, Imperial College, London, UK
[4]NHS England, Leeds, UK

**Correspondence to**
Professor Sophie Staniszewska;
Sophie.Staniszewska@warwick.ac.uk

## ABSTRACT

**Objectives** To review the progress of public involvement (PPI) in NIHR (National Institute for Health Research) research, identify barriers and enablers, reflect on the influence of PPI on the wider health research system in the UK and internationally and develop a vision for public involvement in research for 2025. The developing evidence base, growing institutional commitment and public involvement activity highlight its growth as a significant international social movement.

**Design** The 'Breaking Boundaries Review' was commissioned by the Department of Health. An expert advisory panel was convened. Data sources included: an online survey, international evidence sessions, workshop events, open submission of documents and supporting materials and existing systematic reviews. Thematic analysis identified key themes. NVivo was used for data management. The themes informed the report's vision, mission and recommendations, published as 'Going the Extra Mile—Improving the health and the wealth of the nation through public involvement in research'. The Review is now being implemented across the NIHR.

**Results** This paper reports the Review findings, the first of its type internationally. A range of barriers and enablers to progress were identified, including attitudes, resources, infrastructure, training and support and leadership. The importance of evidence to underpin practice and continuous improvement emerged. Co-production was identified as a concept central to strengthening public involvement in the future. The Vision and Mission are supported by four suggested measures of success, reach, refinement, relevance and relationships.

**Conclusions** The NIHR is the first funder of its size and importance globally to review its approach to public involvement. While significant progress has been made, there is a need to consolidate progress and accelerate the spread of effective practice, drawing on evidence. The outcomes of the Review are being implemented across the NIHR. The findings and recommendations have transferability for other organisations, countries and individuals.

## Strengths and limitations of this study

► The National Institute for Health Research (NIHR) is the first funder of its size and importance globally to review its approach to public involvement.
► The breadth of the evidence collected including from patients, carers and the public, NIHR facilities and institutions, other funders and research organisations and international initiatives.
► Evidence-based policy development that is now being implemented.
► Review primarily focused on research activities of the NIHR.
► Further exploration required to assess equivalence of themes in international contexts.

## BACKGROUND

### Introduction

Public involvement is becoming an increasingly important feature of health research, nationally and internationally. Public involvement—as defined by INVOLVE and adopted for the National Institute for Health Research (NIHR) Review in England is undertaken 'with' or 'by' patients or members of the public, rather than 'to', 'about' or 'for' them.[1] It can mean people becoming members of the research team, or part of reference groups, involved in key discussions and decisions, sharing their unique knowledge, expertise and perspective. For example, they may be involved in identifying key research questions, planning study designs, selecting appropriate outcome measures, collecting data, analysing and interpreting data, disseminating and implementing results.[1] This active involvement is different from people participating as passive subjects in clinical trials with little contribution to identifying its need, designing, conducting or interpreting the trial. It also differs from public engagement which creates a dialogue between researchers and the public to improve public awareness and understanding about research.[2] The intention of public involvement is to prioritise and create research that is relevant, acceptable and appropriate from the patient

or public perspective.[3–6] It may be more likely to be implemented, creating greater impact on health and well-being, particularly if patients also have an active role in implementation.[7] It can also help avoid waste in research by ensuring it focuses on issues of importance and benefit for patients,[8] so maximising the potential for democratic accountability to the wider public, who fund a significant proportion of UK research.

Public involvement is growing as a movement in the UK, Canada, Australia, Europe and the USA. For instance, in the USA, the Patient Centred Outcomes Research Institute (PCORI) encourages patients to submit research questions, provide input on funding applications, participate in events and become an ambassador, reflecting many aspects of NIHR activity.[9] The developing evidence base, and growing institutional commitment to public involvement, highlights its growth as an international social movement, gathering strength and creating significant changes in how research is conducted.[10] Public involvement is focusing on how individuals, communities and patient groups can co-produce with researchers and health professionals, knowledge that will underpin their care and treatment. The potential benefits of public involvement in research and on researchers, patients and the wider community have been identified.[4–6] The beneficial impacts of public involvement on research, researchers, patient and communities include the: identification of patient-relevant topics; grounding of studies in the day-to-day reality of patient experience, enhancing the relevance and appropriateness of studies; identification of patient important outcome measures and solving challenges in securing informed consent. For patients and the public benefits include: feeling listened to and empowered; increased confidence and self-worth and enhanced skills for self-management.[4–6] Patients involved in research can also benefit in a number of ways which can improve their experience of care.[11–13] In summary, public involvement has been found to have a significant role to play in improving the effectiveness and efficiency of research[14] and community and patient empowerment are seen as critical elements in helping the NHS meet future challenges.[15]

Nonetheless, in spite of the emerging evidence base for public involvement over the last 20 years, and a noticeable increase in the number of papers published more recently, challenges remain. These include the quality and utility of the evidence base for practice, including poor conceptualisation, varied definitions, limited capture or measurement of progress of public involvement (PPI) impact and relatively few studies looking at later outcomes of PPI in research.[4–6] A significant difficulty is inconsistent reporting of PPI, with studies often providing partial reporting of their aims, methods and results of PPI in their studies, limiting our understanding of them.[16]

In addition, the practice of PPI is not unproblematic and there is still a significant need to attend to the cultural barriers that inhibit PPI from being completely embedded in research. A recently launched International PPI Network is attempting to create significant culture change in the world of research.[10] In addition, we need to acknowledge that PPI is not always a positive experience with negative impacts reported, particularly on the people involved, if carried out poorly.[4 17] In addition, the tokenism that can exist has been highlighted and the narrowness of current PPI models, with few organisations mentioning empowerment or addressing equality and diversity in their involvement strategies.[18] The potential for poor practice and negative impact made it even more important we undertook the Review to find out how far we have progressed and to understand the current barriers as well as the enablers.

### The UK context

In the UK, the NIHR pioneered a strong policy approach to public involvement including high level support from the Chief Medical Officer.[19] It also established an organisational infrastructure and system for its advancement, delivery and support and INVOLVE, the NIHR funded national advisory group for the promotion and advancement of public involvement. The resulting environment has enabled public involvement to flourish and become a strategic priority for NIHR. Professor Dame Sally Davies (Chief Medical Officer) said,

'No matter how complicated the research, or how brilliant the researcher, patients and the public always offer unique, invaluable insights. Their advice when designing, implementing and evaluating research invariably makes studies more effective, more credible and often more cost effective.'[20]

### The need for the Review

After 10 years of the NIHR promoting and advancing public involvement across its growing infrastructure and associated activities, there was a need to review progress within a UK and international context and to develop a vision for the future and to identify cultural and organisational development required to fulfil the vision of public involvement.

This was particularly important because the extent to which policy support for PPI in health research results in any actual influence on health research agendas also remains unclear.[21] In addition, progress has been relatively slow in funders recognising the importance of funding the substantive development of the PPI evidence base, as opposed to funding the practice of PPI as a stream of activity within a study.

As a result, the 'Breaking Boundaries Review' was announced by the Department of Health on 31 March 2014 and reported as 'Going the Extra Mile'[2] a year later. It was the first such Review by the NIHR of its public involvement work and the first of its type internationally. It was designed as an open and collaborative exercise involving patients, the public, other funders and partners nationally and internationally. The Review Group also felt

the need for the policy review to be evidence-informed in examining progress made and in developing a vision for the future.

## Aims of the Review

1. To review progress made in public involvement in research the UK.
2. To develop a vision for public involvement in research for 2025 and set objectives for the NIHR's leadership in public involvement.
3. To identify cultural and organisational development required to fulfil the vision of public involvement as an embedded component of health research in NIHR.

## METHODS

### Review Panel

A Review Panel was established to shape the Review. Members' expertise included research, policy, research management and patient and public involvement. All members of the panel, including the service users were involved in the planning of the Review, design of the survey, analysis and interpretation and in planning the evidence sessions. Three members were service users. A full list of members is provided in (online supplementary appendix 1).

### Ethical aspects

While formal ethical approval was not sought through an NHS ethics committee for this policy review, it was conducted according to Health Research Authority (HRA) principles of good ethical conduct in research which were applied to relevant stages of the Review. Respondents were invited to read an information sheet about the Review before participating. All respondents were assured of anonymity and confidentiality, unless they gave explicit permission to be quoted. Any identifying information was removed from quotes used within the main report and publications. All submissions were stored on the NIHR CLAHRC Northwest London Imperial College computer system in password protected files.

### Collection of evidence, experiences and perspectives

The Panel carefully considered the type of evidence and information required to address the aims of the Review. The intention of this policy review was not to undertake a review of literature but to be informed by key studies and systematic reviews. There were no formal criteria for inclusion. All members of the Review Group were asked to identify key papers they thought were relevant to the Review. Moreover, the expert Panel also recognised the importance of developing a rigorous process of data collection and analysis, to contribute to high quality evidence-informed policy recommendations. However, it was also felt that a wider collection of evidence, experience and perspectives was necessary, in order to adequately address the Review questions and to meet the NIHR's public involvement values and principles. Five key approaches were selected to facilitate the breadth

---

**Box 1   A summary of methods of data collection**

1. Online questionnaire.
2. Audio and video evidence.
3. Document review.
4. International, third sector and industry representatives evidence panel sessions.
5. Workshops, meetings, social media.

---

of evidence collection, nationally and internationally, summarised in box 1.

### Online questionnaire

A survey monkey online questionnaire was developed in collaboration with the Panel to minimise respondent burden and maximise response. Five key questions were posed to respondents. These were felt by the Panel to align with the aims of the Review and its key themes. The survey questions were also made available as a downloadable word document which could be completed electronically or by hand and posted. A purposive sampling strategy was used to identify a wide range of potential respondents, including individuals and organisations, with the intention of maximising variation in response.[19] Individuals and organisations targeted included patients and members of the public, researchers, clinicians, researchers, user-groups, patient organisations, charities and policy makers nationally and internationally. The initial email with the link to the on-line survey was sent to a range of individuals and organisations, who were asked to cascade it to others nationally and internationally. It was also available on the NIHR INVOLVE website. It was not possible to identify a final sample size because the email was cascaded through the public involvement community and within the NIHR.

### Audio and video evidence

Potential respondents to the call for evidence were offered the opportunity to submit evidence in other formats including audio and video, although no respondents opted for this.

### Document review

Key documents including papers from the NIHR INVOLVE bibliography such as key systematic reviews and grey literature were used to underpin the Review. No systematic review was undertaken due to limited resources. Instead Review Group members and respondents provided key papers, reviews and reports to provide appropriate background and ensure the underpinning evidence base was considered.

### International, third sector and industry evidence

In addition to written submissions, the Review Panel requested input from international colleagues, the third sector and pharmaceutical industry. In total, three panels convened, one panel focusing on international perspectives, one focusing on industry views and one focusing

## Box 2 BMJ Open patient and public involvement reporting criteria

• How was the development of the research question and outcome measures informed by patients' priorities, experience and preferences?
*The question was identified by the Review Panel who included patients. Patients had a key role in shaping the review questions, the methods, the interpretation of the data and the formation of key recommendations.*
• How did you involve patients in the design of this study?
*Patients shaped the design of the review, contributing to the design of the methods for data collection. Patients particularly emphasised the importance of qualitative data collection to capture experiences and perspectives.*
• Were patients involved in the recruitment to and conduct of the study?
*Patients were involved with other panel members to identify and recruit participants. The survey link was cascaded through snowball sampling by patients and PPI leads to key contacts and organisations nationally and internationally.*
• How will the results be disseminated to study participants?
*The study findings will be disseminated through multiple channels including publication, meetings, conferences, social media and through the dissemination plan for NIHR to actively implement the recommendations.*
• For randomised controlled trials, was the burden of the intervention assessed by patients themselves?
*Not applicable.*
• Patient advisers should also be thanked in the contributor ship statement/acknowledgements.
*Patient contributors are thanked in the acknowledgment statement.*

on third sector opinion. Participants were selected based on the knowledge of Review Panel members. The role of the panels was to provide perspectives, insights and any relevant information rather than to have an active involvement role. A set of questions were developed by the Review Panel to support discussion with invited experts which focused on the broader impact of NIHR's public involvement strategy, progress in different sectors, perspectives on how successful the NIHR had been, gaps in provision and areas where it had been less successful.

### Workshops, meetings, social media

Members of the Review Panel joined four workshops hosted by the research team undertaking a key NIHR PPI study called RAPPORT[22] in order to gather evidence. Meetings were held in London, Cambridge, Bristol and Newcastle. Social media was used to publicise the Review, generate debate and encourage submission. An additional workshop was conducted with representatives from medical charities hosted by Parkinson's UK in London. The discussions from workshops, meetings and social media provided a wider context for the Review and its final recommendations but they were not included as part of the NVivo analysis.

### Patient and public involvement

Box 2 reports PPI using the BMJ Open criteria and (online supplementary appendix 2) reports PPI using GRIPP2 Short Form.

### Analysis

Data submitted to the Review via the online survey, by email and post was managed using NVivo software for analysis. Thematic analysis was used to identify key themes emerging from the data.[23] Information provided through other methods was not included in this analysis, but rather provided wider context. A particular focus was on identifying common issues and whether narrative patterns emerged across themes and whether any patterns related to the source of the evidence. Once a submission was received, it was logged and given a unique number and saved to the electronic password protected folder on the Imperial system. Initial thematic analysis was conducted by RM to identify recurrent or common themes. This included responses to the Review questions and the submission of any 'open' evidence. A formative summary was developed by RM of emerging themes, which included a high level summary in the context of the volume and sources of evidence. SS, VM and SD checked meaning and interpretation. The emerging themes were discussed with the Panel to check the interpretation of categories and themes. In order to further structure the analysis RM, SS and VM developed the emerging themes into a coding framework. The data were then analysed according to this framework. Development of themes continued until data saturation, the point at which no new major themes are evolving.[23] As the key themes were identified, SD, with RM, SS and VM identified broader conceptual themes which captured core components of the evidence submitted and provided the conceptual underpinning of the future vision and mission. Panel members also drew on the wider evidence which was documented from the discussion with the international, industry and third sector panels, the regional RAPPORT workshops and workshop with medical charities. Two meetings were held with the Review Panel to scrutinise all available evidence, review interpretations of data and prioritise the report themes.

### RESULTS

Eighty-two responses were received from an individual, institutional, organisational or collective perspective with some submissions representing the combined views, with table 1 reporting respondent characteristics. These included submissions from different parts of the NIHR, medical research charities, universities, industry and third sector bodies. A total of 538 people responded to the online survey. Oral evidence sessions were held with colleagues from USA, Denmark, Germany, Canada and Australia.

Key aspects of Review results are reported in this paper, focusing on positive impacts of PPI, barriers to PPI and then explore how PPI can be undertaken differently. Future delivery is considered and the resulting vision and mission are presented.

### NIHR and INVOLVE as positive influences

The evidence indicated that the NIHR's commitment to include the public in research activity has strengthened

**Table 1** Respondent characteristics

| Respondent characteristic | Number | % |
|---|---|---|
| Public (service user/patient/consumer/carer) | 174 | 40 |
| Researcher/academic | 100 | 23 |
| Other research role (eg, research manager, commissioner) | 39 | 9 |
| Voluntary sector | 27 | 6 |
| User researcher | 24 | 6 |
| Public involvement lead/specialist | 52 | 12 |
| Clinician/practitioner/service provider or manager | 11 | 3 |
| Other | 6 | 1 |
| Total | 433 | 100 |
| Unknown | 105 | – |

over the last 10 years and that the presence and activities of INVOLVE has been important in achieving this. In addition, patients and carers reported a range of positive impacts including gaining insight into the research process and learning more about conditions and treatments. They also reported positive relationships with researchers and welcomed the opportunity to gain new experiences, knowledge, skills and contacts. For example:

'It has given me a platform to represent the views of carers and service users in the design and implementation of research. It has given me a role in life as a lifelong carer I have often felt apart from the world of work and have before my PPI work floated without a purpose.' ID 156 Public

Researchers identified a range of positive impacts including changing their research focus to make it more relevant to patients, altering study designs to take account of experience and improved recruitment. Researchers reported feeling more purposeful and connected to the potential beneficiaries of research.

'It has helped to keep my research close to the concerns of service users. Working with service user researchers in designing studies has been important in keeping the research questions and methodology focused on the concerns of those who will ultimately benefit.' ID 332 Researcher/Academic

### Relevance and usefulness of research with public involvement

Respondents including those from third sector organisations reported that involvement could result in researchers being more likely to address issues of relevance to those with direct experience of a condition, treatment and care. Respondents also described aspects of personal transformation such as gaining new knowledge, changing attitudes and adopting different ways of doing things for example,

'It has enabled increased recruitment through access to hard to reach and minority groups. It has ensured that public facing research materials are accessible and understandable for lay people—again, this increases recruitment. It has enabled evaluation of the experience of those participating in health research—and subsequent trial design has improved, again increasing recruitment. It has ensured where possible that research outcomes are disseminated in a timely and accessible way—resulting in a more informed patient population.' ID 91 Public Involvement Lead/Specialist

### BARRIERS TO PUBLIC INVOLVEMENT IN RESEARCH

Respondents identified a range of ongoing barriers to public involvement including public awareness, attitudes, resources, infrastructure, recognition, reward and payment and resources and training.

### Public awareness

Although there was greater awareness of public involvement in research, it was felt that opportunities were not accessible to the wider population. Evidence submitted by those working in public health particularly emphasised the risk of reinforcing inequalities and missing opportunities to improve health in communities with the most to gain.

'I think the whole "public involvement" side of things is very good at the moment. However, the information (online) about it, such as the opportunities available and how to apply, could be simplified'. ID 32 Public

Many commented on the need for a high profile and well-crafted communication campaign to raise awareness of health research and demystify the activity in a way that the general population could engage with;

'People need to know what is out there, how they can get involved and why it's happening. The acronyms, that then need to be spelt out and explained along with the many avenues an opportunity comes from, suddenly gets too difficult to decipher unless you're an academic or a clinician… ID 227 Other

### Resources

Variability in the availability and allocation of resources to support involvement was a common theme. There was frustration that funding to support relationship building and partnership work ahead of preparing funding applications could be difficult to obtain, but was vital in providing an acceptable standard of good involvement practice in the early stages of research design.

### Infrastructure

As public involvement has grown across the NIHR, variation in the infrastructure to support activity has arisen. This raised questions about how infrastructure decisions

are made, what evidence is available about effective models and to what extent public involvement practice across the NIHR and the NHS can be aligned.

'There is far too much duplication, working in silos and re-inventing the wheel. We need to free ourselves up to enable more time and resources for innovation and creativity. This needs to be joined up with academic and NHS public involvement strategies so that patients have one gateway into involvement opportunities and clear signposting from there'. ID 526 Public Involvement Lead/Specialist

### Recognition, reward and payment

Another significant barrier was the issue of recognition, reward, reimbursement and payment. Despite the availability of guidance, local NHS and Higher Education Institutional policies and administrative practices could be problematical which could slow down prompt reimbursement and payment. Current austerity policies added to those challenges. There is a risk that those who get involved are those who can afford the time and money to do so, compounding issues of exclusion.

'Established groups can provide a wide range of support (research design, pre-funding through to dissemination… However, finance for groups such as these is precarious and without sustained and adequate funding it is difficult for groups to continue to develop and expand their contribution despite the increased requirement for PPI if bids are to be successful. Core funding is needed to fund administrative support of the group as well as advertising, outreach work, mentorship and training of current and new members. ID 29e Researcher/Academic

### Training and support

Many respondents commented on the need for training and support for public involvement. There was broad agreement that a basic level of support should be available to anybody who becomes involved and a minimum skill level and knowledge about public involvement should be incorporated into researcher training. It was acknowledged there is still significant development needed to embed PPI into the research culture in terms of training.

'Currently the training provided is basic, to explain what PPI is and help researchers plan how to proceed (I have taught on such workshops). ID 74 Researcher/Academic

'Training early career researchers in good involvement practice would help increase confidence and understanding of public involvement and reduce the likelihood of bad involvement experiences… ID 19e Charity

### Inconsistency in approach

Some respondents identified difficulties of translating evidence of effective PPI into practice and noted the

evolution of ad hoc practice. Many individuals and teams work independently of each other even within the same organisation, institution or region although there are areas where a more collaborative approach is emerging. For some there is a desire to introduce standards while for others a systematic but flexible approach which addresses key elements such as 'why', 'how' and 'who' would be more helpful.

Making all involvement opportunities task specific, time-limited, with clear expectations and guidance on what people should expect from being involved and how their input will be qualified (eg, two-feedback/appraisal process on how people are performing). Providing information on outcomes of previous, relevant research and examples of how PPI was crucial to the effectiveness of the research trial. 'ID 91 Public Involvement Lead/Specialist

While frameworks for planning evaluations exist, the approach to evaluating PPI was varied and inconsistent.

One would be at the start of a study, to plan ahead how to evaluate the impact of PPI on the research, and on the contributors (cf. the PiiAF – Public Involvement Impact Assessment Framework document). The second would be, with other researchers and PPI representatives acting as 'critical friends', to reflect on a study at the end and thus to work out what to do better next time. ID 74 Researcher/Academic

Some respondents highlighted increasing pressure to demonstrate the impact of PPI and ensure it forms part of a University submission to the Research Excellence Framework, the evaluation of research quality in England and Wales.

### Leadership

A supportive, competent and influential leadership was perceived as critical to the successful delivery of involvement. Respondents commented on the value of experiential knowledge of public involvement in leaders. Conversely, perceived lack of first–hand experience of PPI and limited or absent empathy with patients were thought to diminish the status of some research leaders. It was suggested that champions of involvement are required from outside established involvement teams to promote changes in organisational and institutional culture.

### Challenges

A number of respondents reported poor experiences with PPI including a general sense of frustration from engaging with research, understanding the NIHR and how it links to services. There was also confusion around how to access information and opportunities about becoming involved, suggesting a varied picture of personal practice, organisational commitment and institutional culture.

'I wholeheartedly agree with the intentions and principles of PPI… Unfortunately, I think that lip service is given to PPI by some academics. There is a lack

of transparency about how service users who are involved in research studies are selected, approached, recruited and what biases might be operating.' ID 15 Researcher/Academic

…Some organisations are in a frenzy of PPI because they know they have to do it not because they want to. ID 260 Public

### Scepticism, professionalisation and confusion

Respondents reported a range of challenges when they undertook public involvement, including scepticism about its value, uncertainty about its underpinning theoretical concepts and unclear practice standards. Challenges also included individuals feeling confusion, apprehension and anxiety about how to conduct involvement in a way that demonstrated a positive impact. Researchers were sometimes wary of using experienced advisers because they perceived that the very experience those individuals started from may evolve and be diluted over time. Others felt the development of such specialist expertise was important and had a beneficial impact.

'PPI architecture tends to call for a small number of individuals to make a massive commitment. This means it is hard to find people who can do it and those who do come forward are probably not representative of the wider population. We should try to design more distributed systems which are less clunky and more dynamic (more 'Web 2.0'). Instead of periodic half-day meetings, break things up into smaller modules/components that can be distributed among more people so it is less of a burden for each person. This could allow more people to get involved and it would democratize PPI.' ID 216 Public Involvement Lead/Specialist

While a range of barriers were identified and challenges were identified, respondents recognised that progress in developing and embedding PPI across the NIHR had been made. This had raised the profile of public involvement, established aspects of good practice and made a difference for patients and their families by ensuring research was more meaningful and focused on improved outcomes.

## DOING PUBLIC INVOLVEMENT DIFFERENTLY

Respondents identified new ways of approaching involvement, reflecting a broad range of experience now emerging across the NIHR. A number of key areas for future development emerged.

### Practice standards

There was a perceived need to consolidate and use the available evidence to identify gaps in knowledge. The use of continuous improvement was suggested as way to improve practice standards alongside peer review, performance management, self-regulation and independent regulation. The practice standards have now been launched by INVOLVE and were developed through a consultation process. They provide important guidance and form the basis for continuous improvement.

### Promotion and outreach

Some respondents expressed a desire to extend and deepen the wider involvement of the general population in health research.

'The sense that getting involved in medical research is an aspect of being a good citizen. I think we should foster a sense that the public have a right to participate and, at a minimum level, perhaps even a duty…I think we should build a sense of reciprocity. The public help by volunteering for trials so what does the public get back? …The public pays the going rate for the medicines via the tax system and the NHS. …So I think the reciprocity should come in the form of a bigger say in the direction and shaping of research. ID 216 Public Involvement Specialist/Lead

### Diversity and inclusion

Current involvement practice was perceived by some as being exclusive and not always fully meeting the requirements and goals of equality legislation. Respondents suggested a range of improvements:

'Shorter interactive and more accessible involvement so that everyone can join'. ID 525 Young People Advisory Group Researcher Adviser

'This is difficult for many organisations. Seeing role models like themselves—old/young, non-white, not wearing grey suits—all these would help. People from unrepresented areas may believe that it's not for the likes of them to get involved so showing people who are like them, getting on and making a difference, is likely to be helpful.' ID 29 Public

Respondents also commented on the need for involvement to more closely reflect diversity in the population. It was felt that if leaders and role models were promoted and recruited from varied backgrounds, this would encourage more people to become involved.

'Be more aware of community centres, faith centres as sources of research participants. Acknowledge public health expertise in their local communities; community support officers etc. Get Healthwatch involved. Local radio stations (eg, we have had health/health research message put over local Punjabi radio) Research in the evenings? Weekends? Think differently about when research is done and where it is done. Think who are we going to get participating at that time? The times are usually convenient for the researchers rather than the participants. Make it clear that research studies welcome those with access and mobility difficulties.' ID 240 Other

## THE FUTURE DESIGN AND DELIVERY OF PUBLIC INVOLVEMENT IN NIHR

### Coordinate and collaborate

The NIHR was seen as a complex network of organisations that could benefit from a shared aim for PPI that underpins and informs the development of national policy supported by local practice. Some regions in England and Wales are already moving to a position where individuals from different organisations and programmes are joining to share knowledge and resources, to enhance their own practice.

'Real progress in PPI will not be achieved without an effective mechanism for coordinating PPI efforts across the now many NIHR bodies that have a role in developing, fostering, or implementing PPI. It is essential there is a central body that will coordinate these efforts and will be responsible for ensuring that gaps do not occur, nor needless duplication.' ID 24e Public

### Flexible evidence-based methods

Some respondents suggested that the methods of involvement should be evaluated for their effectiveness. For example, the common practice of inviting one or two patients to join committees was perceived by some to be of limited value and likely to become less attractive as an approach. Many respondents felt that knowledge of the 'ingredients' of effective involvement needed to be developed.

'Involve in the design and delivery as wide a constituency as possible—those with 'knowledge', 'experience' and 'expertise', but also those who may be able to assist by asking questions, because they have different backgrounds.' ID 23e Researcher/Academic

Better identification of the key points where involvement makes an impact was also regarded as important, particularly in relation to deciding research priorities, funding decisions, and translating findings into real benefits for patients. The need for greater openness and transparency in facilitating conversations with the public was also considered important. This would enable patients or members of the public to identify more collaborative or user-led approaches.

'One of the most widely mentioned 'metrics' of improved Public Involvement (PI) would be a growth in collaborative or user-led research. Suggestions for other specific indicators included: routine PI sections in annual reports and evaluation of PI in NIHR funded research project reports; increased representation of people from minority groups; and better recruitment to trials (the latter two suggestions being offered by public contributors).' ID 15e RDS collective

Third sector representatives and community voluntary organisations were identified as potential partners who could more effectively engage with people locally and nationally;

'The voluntary sector could play a key role in both the design and delivery of NIHR funded research. NIHR could establish much stronger links between research charities (such as the Wellcome Trust, Cancer Research UK, the McPin Foundation) and NIHR funded bodies in order to jointly commission and fund research.' ID 35e Voluntary Sector.

### Continuous improvement

Respondents felt there was a need to collect data to enable continuous improvement and not just performance management.

'What is required now is a national framework which sets minimum standards for PPI quality, against which funding and ethical approval decision making can be made. There should also be a move towards making incorporation of quality PPI work into funding application bids standard for all reviewing bodies (as done by NIHR).' ID 51e Other

## DEVELOPING A FUTURE VISION

Many respondents, while recognising progress made so far, expressed the desire to be ambitious for the future. For some, this meant refining practice. For others, it was much more about reframing the purpose of involvement entirely, working differently and recognising the positive connections between engagement, involvement and participation.

### Valued practice

Respondents felt that the debates about the need for public involvement should mature into discussions about what forms of involvement work in particular contexts. Individuals wanted to place their focus on improving the quality of their PPI in creating relevant research.

'PPI should be routine—how things are done, not an optional extra. This should be embedded throughout the NHS so that all users of NHS services can expect that research evidence (is) supported by robust PPI. PPI isn't simply an issue for research but for patient care, too.' ID 15e RDS collective

'By ten years, public involvement should have a much greater profile than what it has now. Members of the public and patients should know that we actively do research in an array of disease areas or conditions and that there are many opportunities for them to take part in this.' ID 20e Public Involvement Lead/ Specialist

### Better evaluation and evidence

The importance of evidence was a key theme, particularly in relation to how to best evaluate public involvement and embedding it into research thinking and practice:

'The evidence base would be substantially enhanced so that there was a consensus between NIHR, senior researchers, the public and other stakeholders on the value of public involvement and the key factors necessary to ensure effective involvement. We will have an agreed set of methods and indicators for assessing the impact of public involvement that will have contributed to building a convincing evidence base. Public involvement would be so embedded in the culture of NIHR that new staff or new researchers coming into the field would naturally take on the values and practices of effective public involvement.' ID 40e Researcher/Academic

### Key concepts

Analysis of the themes emerging from the evidence submissions and synthesis of the data and discussion with the Review led to the development of a mission and vision as presented in (online supplementary appendix 3). Three concepts for measuring success were suggested:

► Reach: the extent to which people and communities are engaged, participating and involved in NIHR research including the diversity of this population.

► Refinement and improvement: how public involvement is adding value to research excellence as funded by the NIHR.

► Relevance: the extent to which public priorities for research are reflected in NIHR funding and activities.

► In addition to these three concepts, as the implementation of the recommendations has progressed, a fourth theme has emerged, relationships. This has been recognised as a significant determinant of success in strengthening public involvement.[22]

Underpinning these concepts is support for the principles of co-production as the basis of the NIHR's approach in the future. These draw on the Boyle[24 25] definition which emphasises the importance of developing close collaborations based on valuing people as assets with knowledge; recognising the expertise and perspective people bring to involvement; promoting good relationships and networks; a perception that all people involved can benefit from public involvement; recognising that involvement often involves an exchange of some type; the process of involvement is important and requires facilitation; that there is a need to change some of the professional boundaries that may inhibit more collaborative forms of work.

### Implementing the Review

In addition to the vision and mission, the Review led to a range of recommendations presented in (online supplementary appendix 4), designed to strengthen co-production and collaboration at the heart of research.

These recommendations are now being actively taken forward across the NIHR. 'Going the Extra Mile' was signed off by the Chief Medical Officer, Professor Dame Sally Davies, in September 2015 with an instruction to NIHR leaders, organisations and staff to support its implementation.[19] This position has been supported with the decision by the Department of Health to regularly audit the NIHR's progress in public involvement using the report recommendations as its starting point. Lines of responsibility and accountability for public involvement have been strengthened accordingly.

The INVOLVE Co-ordinating Centre's future work programme reflects the priorities highlighted in the report and is the NIHR's national lead of diversity and inclusion, learning and development and community (incorporating co-production). A national champion for diversity and inclusion has been appointed. The UK continues to be the only country where national government funds and supports an organisation focused on public involvement in research.

INVOLVE, in partnership with the NIHR's Research Design Services (RDS) organisation, is in the process of supporting and developing regional networks to facilitate collaborative working at local and regional level. These will connect with existing fora and partnerships and will reach across traditional research, services boundaries. Work is ongoing to refresh the way in which the NIHR presents its public involvement work beginning with the new corporate website.

A set of national standards designed to improve the quality and consistency of public involvement in research have been launched.[26] These are based on a set of values and principles for public involvement published by INVOLVE in 2015 and a series of workshops to discuss how best to evolve standards that organisations can operationalise and against which progress can be assessed. A number of NIHR organisations will pilot these standards. This work will feed into emerging thinking about current reporting requirements on initiatives and how these can be improved in ways which will best promote continuous improvement. The Review Panel considered and rejected the notion of a formal regulatory regime for public involvement in favour of an approach which supported and encouraged organisations and their staff.

The programme of reform that is now underway is on top of ongoing innovation in public involvement activity generally. The expansion of the Patient Research Ambassadors Initiative (PRAI) across the NHS, the involvement of young people in research, promotion of public contribution to research through its 'OK to Ask' campaign, and growth of the James Lind Alliance Priority Setting Partnerships programme are all flagship initiatives which continue to receive support from the NIHR within the new strategic framework and approach.

### DISCUSSION AND CONCLUSION

Arguably the NIHR is the leading public research funder globally when it comes to the steps it has taken to make public involvement a core principle for how it funds and supports research excellence. It is perhaps inevitable that it should therefore be the first to attempt a review of the size and scale of 'Going the Extra Mile'.

While the main focus was England, its messages have potential relevance for other countries developing their public involvement, reflecting wider societal changes towards a democratisation of research that enhances the quality of research. Public involvement in the NIHR has made significant progress in the last decade, enabled by a strong policy and infrastructure and implemented by a community of practitioners who recognise the value of actively involving patients and the public in research. We acknowledge the Review was limited to some extent by the lack of a formally conducted review of the literature and would recommend this for future policy reviews.

The Review identified a range of barriers including limited awareness of opportunities, lack of diversity, resistant attitudes to involvement, inconsistent levels of resources, systems that work in different ways, patchy training and support and variable organisational implementation. A key finding from the Review is the need for a step change, increasing the rate of change and with a greater focus on embedding public involvement in research culture, so that it becomes 'business as usual'. The NIHR implementation plan is now starting to address this need but its ambition needs to be recognised as the rule and norms of research need to change for involvement to properly flourish. The Review was strengthened by the involvement of the Review Group, including the service users who ensured there was strong PPI in this example of evidence-informed policy development.

While the focus was national, there are important international implications from the Review. Co-ordination and collaboration across organisations, funders and systems nationally and internationally to deliver high quality public involvement is vital. Public involvement needs to be underpinned by a strong evidence base which enables the development of effective practice that is continuously improved and creates a positive impact. The promotion of opportunities alongside the creation of greater diversity of individuals involved will help ensure a wide range of voices are heard.

In analysing the evidence gathered for the Review, four new key concepts emerged of importance to the field of PPI; reach, refinement and improvement, relevance and relationships. Relationships was added, in the implementation phase, as an additional key concept, vital to the delivery of the future vision. Reach refers to the extent of involvement, engagement and participation, ensuring diversity among members of the public who become involved. To achieve 'reach' researchers and research may need to work closely with the public to develop new ways of working to ensure diversity and inclusion are embedded within involvement. Relevance is focused on the extent to which public priorities for research are reflected in funding and activities. In an era of limited public funding, there is an ethical imperative to ensure public monies are spent on research that patients feel has most relevance to their lives and the beneficial impact it may create. Relevance also refers to

ensuring the research questions in a study are focused on what is acceptable and appropriate from the patient perspective. Drawing on evidence to refine practice through continuous improvement underpins attempts to develop relevance.

The thematic analysis underpinned the development of the vision of '*Going the Extra Mile*', of a population actively involved in research to improve health and well-being for themselves, their family and their communities. The mission of '*Going the Extra Mile*' is of the public as partners in everything we do to deliver high quality research that improves the health, well-being and wealth of the nation. Underpinning this future mission is the principles of co-production, which emerged as a new and important way of understanding the step change required in public involvement.[24 25] At its heart is the co-production of knowledge and evidence through the creation of ways of working, cultures and systems that support this. From a research perspective, co-production offers a way of constructing 'complete' knowledge that includes all relevant aspects of a concept.

'In recent years an approach to research that embeds active participation by those with experience of the focus of that research has been championed both from the human rights perspective, that people should not be excluded from research that describes and affects their lives, and from a methodological perspective in terms of rigorous research: '… knowledge constructed without the active participation of practitioners can only be partial knowledge.''[27]

We emphasise that co-production emerged from the Review and during the process we were not able to explore the concept fully. This in-depth exploration has been conducted by INVOLVE, drawing on a review of literature to inform the development of guidance on co-production and is now published.[28]

In conclusion, 'Going the Extra Mile' challenges researchers, research and the organisations and institutions that fund and promote it to go further in working alongside citizens. The Rome Declaration on Responsible Research and Innovation in Europe in November 2014 emphasises the need to evolve a more inclusive approach to research.

'Hence, excellence today is about more than ground-breaking discoveries—it includes openness, responsibility and the co-production of knowledge.' p. 1

Our vision for the future is ambitious and may take many years to achieve. At its heart, there is a fundamental reorientation of research, its focus, how it is undertaken and how knowledge is created. As others have said, 'if PPI were a drug, it would be malpractice not to prescribe it'. The benefits of co-production could lead us to a new era in research, one that is focused on the co-production of knowledge that benefits humanity in a new and fundamental way. Our ambition is that the 'Going the Extra

Mile' Review escalates such paradigm shift and contributes to changing the nature and role of research, for the benefit of patients, public and wider society.

**Acknowledgements** We would like to thank the Going the Extra Mile Review Panel (online supplementary appendix 1) who worked very hard to shape the Review and participated in all aspects of it. We would like to thank the patients who formed part of the Panel and other patient advisors. Finally, we would like to thank everyone who participated in the generation of evidence for the Review and shaping its recommendations and conclusions.

**Contributors** SD was chair of the Review Group, led on the development of the strategic vision and mission and recommendations in collaboration with the Review Group. SS was vice-chair and supported SD. RM led on the operational aspects of the Review, including the analysis. VM, SS and SD were involved in analysis. SS led the writing of the analysis paper, drawing on the Going the Extra Mile policy report written by SD and RM in collaboration with the Expert Advisory Group.

**Funding** This work was supported by National Institute for Health Research. SS is part-funded by CLAHRC WM. RM is fully funded by NIHR CLAHRC NWL. The 'Going the Extra Mile' policy review was funded by the National Institute for Health Research. This article presents independent research funded by the National Institute for Health Research (NIHR) under the Collaborations for Leadership in Applied Health Research and Care (CLAHRC) programme for North West London and West Midlands.

**Disclaimer** The views expressed in this publication are those of the author(s) and not necessarily those of the NHS, the NIHR or the Department of Health and Social Care.

**Competing interests** None declared.

**Patient consent** Not required.

**Provenance and peer review** Not commissioned; externally peer reviewed.

**Data sharing statement** No additional data available.

**Author note** Dr. Virginia Minogue was at NHS England until October 2017.

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
