## [Reviewer comments · BMJ Open]

ARTICLE DETAILS

TITLE (PROVISIONAL)	Reviewing progress in public involvement in NIHR research: Developing and implementing a new vision for the future
AUTHORS	Staniszewska, Sophie; Denegri, Simon; Matthews, Rachel; Minogue, Virginia

VERSION 1 – REVIEW

REVIEWER	Ewen Speed Senior Lecturer in Medical Sociology School of Health and Human Sciences University of Essex UK
REVIEW RETURNED	29-Jun-2017

GENERAL COMMENTS	This is an interesting paper which addresses a crucial component of contemporary health research and health policy. I have a number of minor concerns that I would like to raise, but I don't have any concerns that substantively impact upon what I think is an important contribution to the ongoing discussion of the role of PPI in healthcare and research. There are a number of points in the paper that I would like to see developed or expanded. Firstly, it strikes me as somewhat odd, that in a review explicitly about user involvement in research, that there was no involvement of researchers or patients or publics in the design of the survey (p9). The reason given was in order to minimise respondent burden but I would argue that an opportunity was missed to explore, in practice, some of the principles of PPI. There is nothing the authors can be expected to do to address this, given data collection has been completed, but they might reflect on this omission, or offer a more detailed response in terms of their perceived need to limit the respondent burden. Surely given the topic of the research it was imperative that patients and other users were involved. In terms of general comments, I felt that the overall emphasis of the entire piece was slanted towards benefits of PPI in terms of research. This is unproblematic, but the authors could have acknowledged a number of wider debates about processes of involvement, links to debates about evidence informed healthcare, evidence based healthcare, and even perceived crises of evidence based medicine (Greenhalgh, Howick, Maskrey , Evidence based medicine: a movement in crisis? BMJ 2014; 348 :g3725) and so on. In addition there are wider debates about the role of the public in health policy, around issues of participative democracy and so forth. To be clear, it is not problematic that the emphasis is on research benefit, but this really needs to be stated much more explicitly towards the start of the piece. Similarly, there is a large body of work
--

that criticises some elements of PPI (most frequently around concerns of tokenism - see for example a paper by one of the authors - Ocloo, Matthews From tokenism to empowerment: progressing patient and public involvement in healthcare improvement BMJ Qual Saf Published doi: 10.1136/bmjqs-2015-004839). In this 2016 paper, Ocloo and Matthews state that "current models of PPI are too narrow, and few organisations mention empowerment or address equality and diversity in their involvement strategies. These aspects of involvement should receive greater attention, as well as the adoption of models and frameworks that enable power and decision-making to be shared more equitably with patients and the public in designing, planning and co-producing healthcare." Issues around empowerment and tokenism could feature more prominently, i.e. in a substantive way in the review. Whilst it is important to acknowledge the standing of PPI in a UK context, it is also important that these other more critical concerns also continue to be raised, otherwise there is a danger that some elements of tokenism continue to persist.

I also felt that the aims for the project (p7), whilst extremely laudable, were not sufficiently developed in the preceding text, or sufficiently introduced in context, they appear (a touch orhapn-like) without any direct framing to explain or link them to the background information that preceded their appearance in the text. The authors could add a paragraph to directly link the aims, as stated, to the preceding background content.

There is an assumption that progress has been made in the 10 years since the NIHR actively promoted PPI, but there is insufficient evidence presented of what this progress looks like, and how it has impacted upon health research in the UK. For example, on page 25, reference is made to fact that PPI has made significant progress in last decade, it would be really useful if the authors could identify some proxy (or actual) indicators that demonstrate this progress, substantive claims like this really need to be supported with some degree of evidence, otherwise there is a danger that we perpetuate a myth of progress (see Madden, M. and Speed, E. (2017) Beware Zombies and Unicorns: Towards critical patient and public involvement in health research in a neoliberal context, *Frontiers in Sociology*, DOI: 10.3389/FSOC/2017.00007). It is important that actual successes are evidenced and underlined, such that they can be used to feed directly into developing best practice elsewhere. As such, I felt this was an opportunity missed to offer up some very real concrete examples of success (this also relates to later point about under-specified practice standards).

Relatedly, on page 9 authors describe how review group members and respondents provided 'key papers, reviews and reports' to 'ensure underpinning evidence was considered' but no indication is given of any of the criteria used to determine key papers from other literature. How were these key papers selected? In addition, there is a lack of clarity for me on how patients were involved in the review. Section 4 makes mention of international, 3rd sector and industry evidence, but it seems to me that patients and publics should also be a group here. Were patients and publics included within these bigger groups? If so, why was there not felt to be a need to differentiate patients into a separate group? These groups, as identified by authors, appear to be deemed 'users' of PPI research. If that is the case, then this raises a concern for me that the role of patients or publics as potential users of PPI evidence is not always

immediately clear from the paper. More clarity on the role of patients and publics across these different groups would be useful.

There is a tendency to understate the evidence underpinning the results of the review. For example, on p17, reference to made to number of respondents reporting poor experiences with PPI, but no differentiation is made between which types of respondents. What sort of proportion of respondents across researchers, patients, public etc reported these issues, was there an over-representation of any particular group in this category? The paper would benefit from presentation of more empirical evidence to support claims made. Similarly, in terms of the issue of understatement, there was some mention of practice standards, but no discussion of what these might look like, or more importantly, what sort of evidence would be used to inform what they might look like. Perhaps this is another paper (and that would be a fair point) but given the authors introduce it, it would be useful to have a bit more detail on the sources of evidence that might be used to formulate these standards (even if this evidence is not currently available in context of this review). For example, continuous improvement was suggested as a way of improving standards, but no indication of what continuous improvement might actually mean, in practice, is offered. What patients regard as continuous improvement might differ (significantly) from what researchers or industry figures might regard as improvement, but little of this complexity is addressed. This resonates with an concern I have that there is also an assumption that PPI works in the same way for all actors, when I would argue there is far more complexity across how PPI works, offering different things for different actors. Different actors have different expectations and different motivations for incorporating PPI into the research process.

Towards the end of the paper, the authors raise the notion of co-production. This is an interesting development (and speaks to some of the concerns I raised about PPI being seen as the same for everyone). Co-production would work to make this understanding much more reflective of the stakes and involvement of the different actors across the PPI spectrum. So this is a positive addition. However, having said that, the placement of co-production as a late addition to the review, does not, in my mind, offer sufficient room for consideration of co-production in relation to PPI. I am somewhat perplexed as to why it is only introduced at this relatively late juncture, as it could have been introduced much earlier, and developed as a key pillar for PPI development across the review. In addition, the concept of co-production is far from any form of consensus. The version cited, (Boyle, 2010) is one approach, amongst others. More detail about Boyle's approach could be offered, as well as an indication that other approaches are available (see Glynos, J. and Speed, E. (2012) Varieties of Co-Production in Public Services: Time Banks in a UK Health Policy Context, *Critical Policy Studies*, 6(4), 402-433, DOI:10.1080/19460171.2012.730760). Given that co-production is stated as underpinning the 4 key concepts of reach, refinement and improvement, relevance and relationships, then to my mind it would significantly improve the paper if the idea of co-production was introduced much more early in the article, and given more discussion and consideration in the text. This also speaks to wider issues in and around notions of participative democracy which are key constituent components of the push and drive for PPI, which could perhaps be more specifically developed across the paper.

	I think this is an important review and provides timely and relevant insight into a very important topic. I hope that the authors feel able to engage with these points in the spirit of constructive criticism that they are offered.
--	--

REVIEWER	Annette Boaz Kingston University and St George's University of London UK
REVIEW RETURNED	29-Jun-2017

GENERAL COMMENTS	This is an interesting and well written report by authors who are extremely knowledgeable about the topic and well known in the field. However, in its current form it doesn't have many of the aspects that might be expected of an academic research paper. It is a largely descriptive review of the progress of PPI in the UK NIHR. It reads like a thorough and well evidenced report, but I couldn't identify any novel contribution to the field or the existing literature or theory. To develop the report further as an academic paper, the analysis would need to move beyond the confines of the NIHR PPI work. One more specific comment, I would have liked to see the longer time frame of PPI in the UK acknowledged. I don't know if the authors are aware of this paper: Evans, D. (2014) Patient and public involvement in research in the English NHS: A documentary analysis of the complex interplay of evidence and policy. Evidence and Policy, 10 (3). pp. 361-377. ISSN 1744-2648 Available from: http://eprints.uwe.ac.uk/21717
--

REVIEWER	Dr Sarah Markham King's College London UK
REVIEW RETURNED	15-Jul-2017

GENERAL COMMENTS	I was immensely cheered to read this review as it provides a much needed overview of the progress of PPI in NIHR research, identifying barriers and enablers and, reflecting on the influence of PPI on the wider health research system in the UK and internationally. The inclusion of direct quotes in the results section provides a wealth of feedback, both positive and pragmatic. This significantly enhances the transparency and relevance of the paper. The finding from the Review of the need for a step change, increasing the rate of change and with a greater focus on embedding public involvement in research culture, so that it becomes 'business as usual', resonates with my own direct experience of PPI in health care research. The two appendices; vision, mission, strategic goals and principles for 2025 and going the extra mile recommendations provide much practical guidance for all healthcare and other organisations which would benefit from PPI. It is a privilege to have reviewed this report and I sincerely hope that it is disseminated widely and its recommendations implemented.
---

VERSION 1 – AUTHOR RESPONSE

Author response

Thank you to all the reviewers for their helpful comments which we believe have significantly strengthened the paper.

Reviewer: 1

Reviewer Name
Ewen Speed

Reviewer 1	Author response
This is an interesting paper which addresses a crucial component of contemporary health research and health policy. I have a number of minor concerns that I would like to raise, but I don't have any concerns that substantively impact upon what I think is an important contribution to the ongoing discussion of the role of PPI in healthcare and research.	Thank you
There are a number of points in the paper that I would like to see developed or expanded. Firstly, it strikes me as somewhat odd, that in a review explicitly about user involvement in research, that there was no involvement of researchers or patients or publics in the design of the survey (p9). The reason given was in order to minimise respondent burden but I would argue that an opportunity was missed to explore, in practice, some of the principles of PPI. There is nothing the authors can be expected to do to address this, given data collection has been completed, but they might reflect on this omission, or offer a more detailed response in terms of their perceived need to limit the respondent burden. Surely given the topic of the research it was imperative that patients and other users were involved.	There was patient involvement in the design of the survey through the involvement of key members of the review group who were patients. We have changed the paper to acknowledge their input. We have also added a reflection at the end of the paper that notes the importance of strong PPI in evidence-informed policy development.
In terms of general comments, I felt that the overall emphasis of the entire piece was slanted towards benefits of PPI in terms of research. This is unproblematic, but the authors could have acknowledged a number of wider debates about processes of involvement, links to debates about evidence informed healthcare, evidence based healthcare, and even perceived crises of	We have added further depth about the negative impacts. We have also noted that PPI is not unproblematic and there is still a significant need to attend the cultural barriers that inhibit PPI. We have referred to: Staniszevska S, Mockford C, Gibson A, Herron-Marx S, Putz R (2011). Moving forward: understanding the negative experiences and impacts of patient and public

evidence based medicine (Greenhalgh, Howick, Maskrey , Evidence based medicine: a movement in crisis? BMJ 2014; 348 :g3725) and so on. In addition there are wider debates about the role of the public in health policy, around issues of participative democracy and so forth. To be clear, it is not problematic that the emphasis is on research benefit, but this really needs to be stated much more explicitly towards the start of the piece. Similarly, there is a large body of work that criticises some elements of PPI (most frequently around concerns of tokenism - see for example a paper by one of the authors - Ocloo, Matthews From tokenism to empowerment: progressing patient and public involvement in healthcare improvement BMJ Qual Saf Published doi: 10.1136/bmjqs-2015-004839). In this 2016 paper, Ocloo and Matthews state that "current models of PPI are too narrow, and few organisations mention empowerment or address equality and diversity in their involvement strategies. These aspects of involvement should receive greater attention, as well as the adoption of models and frameworks that enable power and decision-making to be shared more equitably with patients and the public in designing, planning and co-producing healthcare." Issues around empowerment and tokenism could feature more prominently, i.e. in an substantive way in the review. Whilst it is important to acknowledge the standing of PPI in a UK context, it is also important that these other more critical concerns also continue to be raised, otherwise there is a danger that some elements of tokenism continue to persist.	involvement in health service planning, development and evaluation. In: Barnes, Marian and Cotterell, Phil eds. Critical Perspectives on User Involvement. Bristol: The Policy Press, pp. 129–141. We have reviewed the content of the paper in relation to the comment on wider debates about participative democracy and how PPI is linked to a wider movement. In the submitted paper we already refer to wider social movements so we have not added to this section, mindful of additions elsewhere and the limited word count. We have referred to the Ocloo paper particularly as Matthews is one of the co-authors on both papers, highlighting the challenges of narrow models that do not consider diversity and equality. Please note diversity and inclusion is one of the key themes in the results section and this is also addressed in the recommendations in relation to our community being diverse.
I also felt that the aims for the project (p7), whilst extremely laudable, were not sufficiently developed in the preceding text, or sufficiently introduced in context, they appear (a touch orphan-like) without any direct framing to explain or link them to the background information that preceded their appearance in the text. The authors could add a paragraph to directly link the aims, as stated, to the preceding background content.	We have added new sentences in the section 'the need for the review' to link the need with the aims more specifically.
There is an assumption that progress has been made in the 10 years since the NIHR actively promoted PPI, but there is insufficient evidence presented of what this progress looks like, and	Thank you for this comment. In many ways it provides the rationale for why we undertook the review, to assess perceptions and examples of progress. We agree with the reviewer that

how it has impacted upon health research in the UK. For example, on page 25, reference is made to fact that PPI has made significant progress in last decade, it would be really useful if the authors could identify some proxy (or actual) indicators that demonstrate this progress, substantive claims like this really need to be supported with some degree of evidence, otherwise there is a danger that we perpetuate a myth of progress (see Madden, M. and Speed, E. (2017) Beware Zombies and Unicorns: Towards critical patient and public involvement in health research in a neoliberal context, Frontiers in Sociology, DOI: 10.3389/FSOC/2017.00007). It is important that actual successes are evidenced and underlined, such that they can be used to feed directly into developing best practice elsewhere. As such, I felt this was an opportunity missed to offer up some very real concrete examples of success (this also relates to later point about under-specified practice standards).	perhaps our introductory sections sounded a little optimistic so have tempered them. We have linked the intention of the Review to the desire to better understand the barriers as well as the enablers. The reviewer is correct to question progress. We have already referred to key systematic reviews and noted the nature of impacts made on individuals, patients, researchers, communities and research. So it's important we recognise the impacts that have been made. We have perhaps added more notes of caution of wider impacts on research agenda, particularly drawing on Madden and Morley (2016). We agree there is still progress in many aspects of research culture. We have noted that progress has been slow in relation to developing the evidence base of PPI, with funders now always recognising the importance of evidence-based practice. We have also noted the need for significant cultural change in research which is difficult to achieve and requires cross-country collaboration. We highlight that this has been noted recently and an international network is beginning to form in recognition of the slow progress of cultural change.
Relatedly, on page 9 authors describe how review group members and respondents provided 'key papers, reviews and reports' to 'ensure underpinning evidence was considered' but no indication is given of any of the criteria used to determine key papers from other literature. How were these key papers selected?	The intention of this policy review was not to undertake a review of literature but to be informed by key studies and systematic reviews. We have added this clarity. All members of the Review Group were asked to identify key papers they thought were relevant to the Review. There were no formal criteria for inclusion. The focus of this Review was on reaching out to the wider PPI community rather than on the basis of the

	literature, hence the focus on being informed by key papers. We have noted the limitations of this in the discussion and recommended that if resources allow future policy reviews might consider a more formal literature review process.
In addition, there is a lack of clarity for me on how patients were involved in the review. Section 4 makes mention of international, 3rd sector and industry evidence, but it seems to me that patients and publics should also be a group here. Were patients and publics included within these bigger groups? If so, why was there not felt to be a need to differentiate patients into a separate group? These groups, as identified by authors, appear to be deemed 'users' of PPI research. If that is the case, then this raises a concern for me that the role of patients or publics as potential users of PPI evidence is not always immediately clear from the paper. More clarity on the role of patients and publics across these different groups would be useful.	Patients were included in the main Review Group (3) so were embedded in the process, influencing each aspect. Section 4 – we have added clarity to state that participants on the panels were selected based on the knowledge of Review panel members. The panels of international, 3rd sector and industry included patients but the role of the panels was to provide perspectives, insights and any relevant information rather than to have an active involvement role. We include named members of the Review group in the acknowledgments.
There is a tendency to understate the evidence underpinning the results of the review. For example, on p17, reference to made to number of respondents reporting poor experiences with PPI, but no differentiation is made between which types of respondents. What sort of proportion of respondents across researchers, patients, public etc reported these issues, was there an over-representation of any particular group in this category? The paper would benefit from presentation of more empirical evidence to support claims made.	The intention of the review was to provide a qualitative insight into the range of experiences of involvement. The intention wasn't to provide a quantitative evaluation of issues based on different groups.
Similarly, in terms of the issue of understatement, there was some mention of practice standards, but no discussion of what these might look like, or more importantly, what sort of evidence would be used to inform what they might look like. Perhaps this is another paper (and that would be a fair point) but given the authors introduce it, it would be useful to have a bit more detail on the sources of evidence that might be used to formulate these	The reviewer is right to ask for clarity on continuous improvement. The practice standards are now being developed and we have added a couple of lines to the paper to clarify this and to highlight their role in continuous improvement, although the exact plan for this work is currently being formed

standards (even if this evidence is not currently available in context of this review). For example, continuous improvement was suggested as a way of improving standards, but no indication of what continuous improvement might actually mean, in practice, is offered. What patients regard as continuous improvement might differ (significantly) from what researchers or industry figures might regard as improvement, but little of this complexity is addressed. This resonates with an concern I have that there is also an assumption that PPI works in the same way for all actors, when I would argue there is far more complexity across how PPI works, offering different things for different actors. Different actors have different expectations and different motivations for incorporating PPI into the research process.	by NIHR INVOLVE. The standards work may merit their own paper and we have not expanded on their development as this is a separate piece of work being led by INVOLVE.
Towards the end of the paper, the authors raise the notion of co-production. This is an interesting development (and speaks to some of the concerns I raised about PPI being seen as the same for everyone). Co-production would work to make this understanding much more reflective of the stakes and involvement of the different actors across the PPI spectrum. So this is a positive addition. However, having said that, the placement of co-production as a late addition to the review, does not, in my mind, offer sufficient room for consideration of co-production in relation to PPI. I am somewhat perplexed as to why it is only introduced at this relatively late juncture, as it could have been introduced much earlier, and developed as a key pillar for PPI development across the review. In addition, the concept of co-production is far from any form of consensus. The version cited, (Boyle, 2010) is one approach, amongst others. More detail about Boyle's approach could be offered, as well as an indication that other approaches are available (see Glynos, J. and Speed, E. (2012) Varieties of Co-Production in Public Services: Time Banks in a UK Health Policy Context, Critical Policy Studies, 6(4), 402-433, DOI:10.1080/19460171.2012.730760).	Thank you to the reviewer for their favourable response to this. We agree it is very important. The concept emerged towards the end of the Review rather than at the start to we were not able to use it to shape the Review process. We have added text to clarify this. We would acknowledge that the paper does not offer sufficient room to explore this in depth. We have added text to explain this and to note that there is a separate piece of work underway to explore how co-production will work within the context of NIHR. This will address different definitions and approaches. Boyle is used in the context of this paper as a means of introducing the concept rather than as a definitive view on co-production. We have added text to make this clear.
Given that co-production is stated as underpinning the 4 key concepts of reach, refinement and improvement, relevance and relationships, then to my mind it would significantly improve the paper if the idea of co-	Thank you. This is a helpful point. We introduced co-production late in the paper because it emerged from the Review as the core concept, so to introduce it earlier might break the flow of the paper and introduce this result

production was introduced much more early in the article, and given more discussion and consideration in the text. This also speaks to wider issues in and around notions of participative democracy which are key constituent components of the push and drive for PPI, which could perhaps be more specifically developed across the paper.	too soon. We feel the current work on developing our understanding of co-production being led by INVOLVE will create the opportunity for further clarity in a separate paper. While we agree with the comment about participative democracy, we but are not able to expand on this due to restricted word count. If the Editor would like us to we would be more than happy, but it does take us slightly outside of the remit of the Review which was conducted at the request of a funder.
I think this is an important review and provides timely and relevant insight into a very important topic. I hope that the authors feel able to engage with these points in the spirit of constructive criticism that they are offered.	Thank you. We very much appreciate the time the reviewer has put into very careful review of our paper with some incredibly useful suggestions.
Reviewer 2	
Please leave your comments for the authors below This is an interesting and well written report by authors who are extremely knowledgeable about the topic and well known in the field. However, in its current form it doesn't have many of the aspects that might be expected of an academic research paper. It is a largely descriptive review of the progress of PPI in the UK NIHR. It reads like a thorough and well evidenced report, but I couldn't identify any novel contribution to the field or the existing literature or theory. To develop the report further as an academic paper, the analysis would need to move beyond the confines of the NIHR PPI work.	We agree that the paper provides a descriptive review of the current status of the field. We do however acknowledge that this paper is a review of policy and practice, rather than the more traditional research paper, hence we understand why the reviewer feels some elements are missing. However, we feel this type of paper is of value and no other paper has provided such an overview based on such a large survey. We anticipate significant international interest, particularly from countries who are trying to establish PPI strategy and practice. The key concepts that emerge are the focus on co-production which has not been identified in the same way within more conventional forms of research. We have tried to highlight this. In addition the new concepts of reach, relevance, refinement and relationships are new to this area of policy. We have highlighted these more specifically as new contributions.
One more specific comment, I would have liked to see the longer time frame of PPI in the UK acknowledged. I don't know if the authors are aware of this paper:	Thank you.

Evans, D. (2014) Patient and public involvement in research in the English NHS: A documentary analysis of the complex interplay of evidence and policy. Evidence and Policy, 10 (3). pp. 361-377. ISSN 1744-2648 Available from: http://eprints.uwe.ac.uk/21717	
Reviewer: 3	
I was immensely cheered to read this review as it provides a much needed overview of the progress of PPI in NIHR research, identifying barriers and enablers and, reflecting on the influence of PPI on the wider health research system in the UK and internationally. The inclusion of direct quotes in the results section provides a wealth of feedback, both positive and pragmatic. This significantly enhances the transparency and relevance of the paper. The finding from the Review of the need for a step change, increasing the rate of change and with a greater focus on embedding public involvement in research culture, so that it becomes 'business as usual', resonates with my own direct experience of PPI in health care research. The two appendices; vision, mission, strategic goals and principles for 2025 and going the extra mile recommendations provide much practical guidance for all healthcare and other organisations which would benefit from PPI. It is a privilege to have reviewed this report and I sincerely hope that it is disseminated widely and its recommendations implemented.	Thank you. It is heartening to read your comments.